# Toxicological Evaluation Verifies the Safety of Oral Administration of Steamed Mature Silkworm Powder in Rats

**DOI:** 10.3390/foods13142209

**Published:** 2024-07-13

**Authors:** Young-Min Han, Da-Young Lee, Moon-Young Song, Eun-Hee Kim

**Affiliations:** College of Pharmacy and Institute of Pharmaceutical Sciences, CHA University, Seongnam 13488, Republic of Korea; han.ymin3@gmail.com (Y.-M.H.); angela8804@naver.com (D.-Y.L.); wso219@naver.com (M.-Y.S.)

**Keywords:** steamed and freeze-dried silkworm larva powder, edible insect, oral toxicity, repeat administration toxicity, NOAEL

## Abstract

Globally, the edible insect industry is emerging due to its potential contributions to food security and environmental sustainability. Edible insects are rapidly being integrated into the development of alternative foods and new pharmaceuticals. Silkworms, known for their high protein content, are not only a potential new source of human food and animal feed but have also been traditionally used for medicinal purposes. However, conventional silkworms are difficult to ingest. To address this, we have developed a steamed and freeze-dried mature silkworm larva powder (SMSP), and it is essential to investigate its potential toxicity and food safety for further studies and applications. Therefore, this study aimed to evaluate the toxicity of SMSP. A toxicity assessment of SMSP was conducted according to OECD guidelines. An oral repeat-administration study was performed on male and female SD rats at doses of 625, 1250, and 2500 mg/kg/day for 4 and 13 weeks. No toxicological changes were observed in clinical signs, body weight, water and food intake, urine tests, hematology, clinical biochemistry, gross findings, or histopathological examination. In conclusion, the no observed adverse effect level (NOAEL) of SMSP was 2500 mg/kg/day, with no target organs identified in either sex of the rats. These results suggest that SMSP is safe, is without side effects and has potential for use as an edible ingredient and in health functional food applications.

## 1. Introduction

The consumption of insects has garnered global attention due to its health, environmental, and economic benefits [1]. Edible insects, abundant in protein, fat, minerals, vitamins, and fiber, can play a crucial role in addressing food insecurity [1]. The benefits of incorporating insects into the diet include high feed conversion efficiency and the ability to rear insects on organic waste, thereby mitigating environmental pollution [2]. Insects produce fewer greenhouse gases and require considerably less land and water compared to traditional livestock [3]. Additionally, the nutritional quality of insects can be equivalent to or even exceed that of foods derived from birds and mammals [4]. For this reason, Europe and the United States are rapidly incorporating insects into the development of alternative foods and new pharmaceuticals [5]. Despite these benefits, the utilization of edible insects faces challenges, including the need for standardization, quality control, and food safety regulations [1,6]. Addressing these issues could establish edible insects as a sustainable major food source, helping to meet global food demands amid population growth and climate change [6].

Silkworms, particularly popular in Asia, see an increase in silk protein during larval growth, with a significant rise after the third day of the fifth instar as they become filled with silk fibers and their silk glands enlarge [7]. Numerous studies have confirmed that silkworms are rich in various nutritional ingredients, making them suitable for health functional foods [8]. Silkworms, the larvae of *Bombyx mori*, are rich in silk proteins, primarily fibroin and sericin [9], and they also contain vitamins, minerals, and omega-3 polyunsaturated fatty acids (ω-3 PUFA) [10]. Historically, silkworms have been used in traditional medicine to treat conditions such as flatulence, phlegm, and seizures [11]. According to recent studies, silkworms have been shown to provide significant health benefits, including positive effects on liver disease [12], a pancreatic protective effect [13], an anti-hypertensive effect [14], a hypoglycemic effect [15,16], an anti-atopic dermatitis effect [17], a preventive effect on memory impairment [18], and antioxidant activity [19]. Despite these benefits, silkworms present ingestion challenges due to their high silk protein content, which limits their utilization and commercial availability. To address this issue, our collaborators developed a method involving steaming mature larvae at 100 °C for 130 min, followed by lyophilization and pulverization [7]. This process yielded Silkworm Mature larvae Steam Powder (SMSP), which contains higher levels of proteins, amino acids, essential minerals and nutrients, including omega-3, compared to the conventional third day fifth instar silkworm powder [7]. When comparing non-steamed silkworm with SMSP, the crude protein content was 68.7% for steamed silkworms and 62.4% for non-steamed silkworms, indicating a significant difference [7]. Additionally, the levels of serine (1.1 times), glycine (1.3 times), alanine (1.3 times), and tyrosine (1.2 times) were higher in SMSP than in non-steamed silkworm [7]. There were no differences in fatty acids, vitamins, dietary fiber, or moisture content, but the protein content, making up 69~72% of the general components of SMSP, significantly contributes to its functionality [7]. Therefore, toxicity tests using SMSP are essential to assess whether such a high protein intake could affect kidney and liver function or cause other disorders. Currently, there is no international legislation specifically regulating the use of silkworm powder in food products. This study aims to provide comprehensive data supporting the safety of silkworm powder for consumption, contributing valuable information to the field and potentially aiding in the establishment of future regulations.

Recently, our previous study reported that SMSP has a preventive effect on ethanol-induced gastric ulcers and reduces pro-inflammatory mediators in alcohol-induced gastric injury using Sprague-Dawley (SD) rats [20]. Furthermore, we found that SMSP intake was effective in reducing ethanol-induced fat accumulation, lowering the low-density lipoprotein/high-density lipoprotein ratio, and restoring overall antioxidant levels in animal experiments with alcohol-induced fatty liver disease [12]. Additionally, our previous study has shown that SMSP mitigates hepatocellular carcinogenesis and hepatic fibrosis by inhibiting transforming growth factor (TGF)-β activity and the phosphorylation of signal transducer and activator of transcription (STAT)-3 signaling in diethylnitrosamine-induced hepatocellular carcinoma rat models [21]. However, despite these positive effects, there is a notable absence of safety studies on SMSP. Consequently, there is a need for comprehensive research to evaluate the potential safety issues associated with SMSP. The maximum daily intake allowance of SMSP can be determined based on the results of toxicity tests. To date, no adverse effects from silkworm consumption in humans have been reported. Therefore, to widely use SMSP as new food source or health functional food, it is necessary to conduct toxicological safety assessments for long-term exposure. This study aims to systematically evaluate the potential chronic stability issues related to SMSP by conducting 4- and 13-week repeated-dose oral toxicity trials in SD rats, in accordance with OECD guidelines.

## 2. Materials and Methods

### 2.1. Preparation of Extracts

SMSP was prepared according to the method previously described [9]. In brief, mature larvae of the *Bombyx mori* (Baekokjam strain) were cultivated on mulberry leaves at the National Institute of Agricultural Science in Korea. The larvae were immediately subjected to a steaming process at 100 °C for 130 min using a non-electric pressure-cooking appliance (KumSeong Ltd., Bucheon, Republic of Korea). Post-steaming, the larvae were freeze-dried (FDT-8612, Operon Ltd., Gimpo, Republic of Korea) for 24 h. The freeze-dried larvae were then pulverized using a hammer mill (HM001, Korean Pulverizing Machinery Co., Ltd., Incheon, Republic of Korea) followed by a disk mill (Mil101, Korean Pulverizing Machinery Co., Ltd., Incheon, Republic of Korea). The SMSP particles were refined to a size of less than 0.1 mm and subsequently stored at −50 °C. The general composition of SMSP is 68.7% crude protein, 10% crude fat, 3.26% crude ash, 2.85% moisture, and 1.75% crude fiber [7]. For each experimental study, SMSP was freshly prepared by dissolving the powdered extract in distilled water (DW).

### 2.2. Animals and Housing Conditions

A total of 80 five-week-old Specific pathogen Free (SPF) SD rats were obtained from Orient Bio (Seongnam, Republic of Korea). Upon arrival, the animals were subjected to a 6-day acclimatization period prior to the initiation of the experimental procedures. Following acclimatization, the mice were weighed and randomly allocated into four experimental groups. The baseline body weight for the male rats ranged from 193.82 to 217.21 g, and the range for the female rats was 144.436–176.08 g. The animals were housed in an environmentally controlled facility with the following conditions: temperature maintained at 23 ± 3 °C, relative humidity at 55 ± 15%, and ventilation rate of 10–20 air changes per hour. The light/dark cycle was set to 12 h (lights on at 8:00 a.m. and off at 8:00 p.m.) with an illuminance of 150–300 Lux. The animals were provided with unrestricted access to standard laboratory food and water. The experimental procedure was reviewed and approved by the Institutional Animal Care and Use Committee (IACUC) of Chemon Inc. (Yongin, Republic of Korea) (approval number: 18-R644).

### 2.3. Experimental Procedure for 13-Week Repeated Oral Dose Toxicity Study

The research adhered to the following testing standards: OECD Guideline for Testing of Chemicals (specifically Guideline 408, which pertains to the ‘Repeated Dose 90-day Oral Toxicity Study in Rodents’). For the 13-week repeated oral dose toxicity study, SMSP was orally administered once daily in the morning for duration of 13 weeks, with dosages set to 625, 1250, and 2500 mg/kg BW/day. The treatment groups, comprising 10 rats for each gender and group, were as follows: vehicle control group, SMSP low-dose group (625 mg/kg BW/day), SMSP middle-dose group (1250 mg/kg BW/day), and SMSP high-dose group (2500 mg/kg BW/day). The dose was administered at a volume of 20 mL/kg/day based on the most recently measured body weight. Clinical signs, including mortality, general appearance, and behavioral abnormalities, were monitored daily in the experimental rats until the point of sacrifice. Additionally, body weight, food consumption, and water intake were recorded on a weekly basis throughout the duration of the study. At study termination, all the rats were euthanized via inhalation of isoflurane (2–5%) to collect blood samples.

#### 2.3.1. Urinalysis

During the last week of the administration period, 5 animals/sex/group were housed in metabolic cages. Urine samples were collected over a 24 h period. The assays included measurements of glucose (GLU), bilirubin (BIL), ketone bodies (KET), protein (PRO), urobilinogen (URO), specific gravity (SG), and nitrite (NIT), and were conducted using an auto-analyzer (Clinitek Advantus, Siemens, Munich, Germany).

#### 2.3.2. Hematology and Biochemical Analysis

Blood samples were collected from the abdominal aorta and utilized for both hematologic parameters and serum biochemical analysis. Following collection in anticoagulant-containing bottles, 14 parameters were analyzed using a Coulter counter (Siemens, Tarrytown, NY, USA): red blood cell (RBC) count, hemoglobin (HGB) level, hematocrit (HCT), mean corpuscular volume (MCV), mean corpuscular hemoglobin (MCH), mean corpuscular hemoglobin concentration (MCHC), platelet (PLT) count, mean platelet volume (MPV), white blood cell (WBC) count, neutrophil (NEU) count, lymphocyte (LYM) count, monocyte (MONO) count, eosinophil (EOS) count, basophil (BASO) count, and large unstained cell (LUC) count. Additionally, 17 serum biochemical parameters were measured using serum isolated immediately from the collected whole blood samples (AU680; Beckman Coulter, CA, USA): aspartate aminotransferase (AST), alanine aminotransferase (ALT), alkaline phosphatase (ALP), creatine phosphokinase (CPK), total bilirubin (TBIL), glucose (GLU), total cholesterol (TCHO), triglycerides (TG), total protein (TP), albumin (ALB), blood urea nitrogen (BUN), creatinine (CRE), inorganic phosphorus (IP), calcium ion (Ca^2+^), sodium ion (Na^+^), potassium ion (K^+^), and chloride ion (Cl^−^).

#### 2.3.3. Gross Findings, Organ Weights, and Histopathological Analysis

Necropsies were conducted on the animals, and detailed findings from the autopsy procedure were documented for all organs. Following the necropsy, the ovaries, cervix, prostate, testicles, epididymis, thymus, lungs, brain, pituitary gland, spleen, liver, heart, kidneys, and adrenal glands were excised, and their absolute weights were measured using an electronic scale (Secura 224-1S; Sartorius AG, Göttingen, Germany). The testes and epididymides were fixed in Bouin’s solution, while the eyes were preserved in Davidson’s solution. All other organs were fixed in a 10% formalin solution. Lesions were graded using a five-step scale ranging from minimal to large-scale severity. Histopathological findings were analyzed using Prestima^®^ (Xybion, Princeton, NJ, USA). Key organs were treated and embedded in paraffin wax to prepare tissue sections, which were then stained with hematoxylin and eosin (HE). The stained slides were examined under an upright microscope.

### 2.4. Statistical Analysis

The results were expressed as the mean ± standard deviation (SD). All statistical analyses were conducted using SPSS Statistics 22 for Medical Science (SPSS), and we adhered to the standard working guidelines for statistical processing in our laboratory. The weight, feed intake, hematological parameters, blood biochemical parameters, and organ weights were analyzed using a one-way analysis of variance (ANOVA). A test for equal variance was performed. If equal variance was observed, the analysis was followed by Duncan’s test. In cases where equal variance was not identified, the Dunnett’s T3 test was utilized.

## 3. Results

### 3.1. Four-Week Repeated-Dose Toxicity Study

To assess toxicity and establish the dosage for a 13-week repeat-administration toxicity test, a study was conducted with SMSP administrated orally at doses of 312.5, 625, 1250, and 2500 mg/kg/day for 4 weeks. The results showed no substance-related changes in mortality, clinical signs, body weight, food and water intake, ophthalmic examinations, urine tests, hematological and clinical biochemical tests, long-term body weight, or visual finding during the 4-week treatment period. Additionally, no significant weight changes were observed during the trial period (Figure 1). It was concluded that repeated oral administration of SMSP for 4 weeks in SD rats did not result in any biologically significant changes. Therefore, the high dose for the 13-week repeat-administration toxicity test was set at 2500 mg/kg/day.

### 3.2. Thirteen-Week Repeated-Dose Toxicity Study

#### 3.2.1. Mortality, Clinical Signs, Ophthalmic Examination, Body Weight and Food Intake

We provided a reasonable basis for dose selection through a 4-week repeated-dose oral administration toxicity test. Therefore, a dose of 2500 mg/kg/day was chosen as the high dose, with medium and low doses set at 2-fold intervals. The rats were orally administered SMSP at doses of 625, 1250, and 2500 mg/kg/day for 13 weeks. During the test period, no animal deaths or other symptoms of toxicity were observed in any treatment group. Although one male rat in the control group died on day 82, no specific symptoms were observed before death. Additionally, several rats exhibited symptoms such as a bite wounds, scratch wounds, and loss of fur. However, these changes were considered incidental and independent of the test substance as they were few in number and showed no dose–response correlation. Ophthalmological examinations revealed no treatment-related abnormalities in any group of either sex. As shown in Figure 2, the weight of the experimental animals in all treatment groups increased progressively throughout the study, regardless of the varying doses of SMSP (Figure 2). No test substance-related changes were observed in body weight, food intake, or water consumption between the vehicle control and test substance treatment groups.

#### 3.2.2. Urinalysis, Hematological Analysis, Clinical Biochemistry Assessment

Table 1 shows the results of analyzing five urine samples from each group, with each number representing the corresponding number of animals. As shown in Table 1, the ketone (KET) levels in the urine were significantly decreased, and the urine volume was increased in the male group receiving 1250 mg/kg/day (*p* < 0.05) (Table 1). However, these changes were not considered treatment-related due to the lack of dose-dependence. Furthermore, the hematological and serum biochemical parameters showed no statistically significant changes in any of the groups exposed to SMSP compared to the control group (Table 2 and Table 3).

#### 3.2.3. Organ Weights, Gross Abnormality, and Histopathological Analysis

During autopsy, when the organ weights were measured, no changes related to the test substance were observed for any organ (Table 4). Additionally, no toxic lesions associated with the test substance were detected in any group during visual examination at autopsy and histopathological analysis. In previous studies, we demonstrated that SMSP has a protective effect on liver disease [12]. Specifically, even upon closer examination of the histopathology of liver tissue, no significant differences were observed between the high-concentration SMSP group and the control group (Figure 3).

## 4. Discussion

Prior to this study, we confirmed the absence of SMSP toxicity in vitro through genotoxicity studies, including bacterial reverse mutation assays, mammalian chromosomal aberration tests, and mammalian erythrocyte micronucleus assays. Subsequently, a 4-week repeated-dose oral administration study was conducted to determine the appropriate dose for a 13-week toxicity test. During the 4-week trial, no signs of toxicity were observed in any test animals at a dose of 2500 mg/kg/day. Therefore, a 13-week repeated-dose oral administration toxicity study was performed at this dose. 

During the test period, we did not observe any animal deaths or overt signs of toxicity in the treatment groups. Although one male rat in the control group died on day 87, no specific symptoms were noted prior to its death, and this occurrence was rare and limited to the control group, suggesting it was likely an incidental issue. Additionally, several rats exhibited symptoms such as bite wounds, cryptorchidism (undescended testis), ear swelling, scratch wounds, crust formation, fur loss, teeth malocclusion, and scars. However, these changes were considered to be accidental and unrelated to the test substance, as they were infrequent and did not show a dose–response relationship. Throughout the experimental period, the animals’ body weight increased naturally, and there were no significant changes in food and water intake. Furthermore, no significant alterations were observed in urinalysis, blood analysis, biochemical blood analysis, organ weight, or pathological assessments attributable to SMSP. Although the urinalysis indicated a significant reduction in the KET level in one male rat in the 1250 mg/kg/day group, no corresponding changes in organ weight or histopathological findings of the kidney and liver were detected. The lack of a dose–response relationship suggests that these changes were toxicologically insignificant and unrelated to the test substance. In this study, SD rats were orally administered SMSP at doses of 625, 1250, and 2500 mg/kg/day for 13 weeks, with no discernible effects observed from the test substance. Therefore, the no observed adverse effect level (NOEAL) of SMSP under these test conditions is suggested to be 2500 mg/kg/day for both sexes, with no specific target organs identified. 

In comparison, a toxicity test using silkworm extract powder (SEP) reported in 2013 established a NOAEL of 2000 mg/kg BW/day. This SEP, derived from fifth instar third day larvae, contained significant amounts of 1-deoxynojirimycin (DNJ), a hypoglycemic component [22]. In contrast, SMSP produced through steaming and grinding mature silkworms contains higher levels of beneficial ingredients such as amino acids and omega-3 fatty acids and demonstrates enhanced safety. However, SMSP has a very low DNJ content, likely due to the absence of silkworm feces in the fully grown silkworms used [23]. Given their different effective ingredients and functions, SEP and SMSP can be applied based on specific needs and have the potential to complement each other in development.

Toxicity studies have also been reported for other insects such as mealworms and grasshoppers as well as silkworm [24,25]. Like silkworms, these insects are gaining recognition as future food resources. Their toxicity and allergic reactions when used as food or dietary supplements have been analyzed. Specifically, in Europe, dried *Tenebrio molitor* larvae [26], freeze-dried and powdered *Locusta migratoria* [27], freeze-dried and powdered *Acheta domesticus* [28], and frozen and freeze-dried lesser mealworm [29] have been approved under food regulations. The European Food Safety Authority (EFSA) conducted rigorous evaluations of their toxicity and allergenicity, confirming their safety for consumption. Increasing research supports the safety of insect consumption, and various insects have been found to be safe as food. Therefore, the toxicity study of SMSP in this research provides crucial scientific evidence for future official certification.

These findings have significant implications for the development of SMSP as a novel functional health food. Specifically, they are expected to pave the way for further research into SMSP’s mechanisms and its effectiveness against various diseases. Our research team has previously reported several positive effects of SMSP, including improvements in liver function [12,21]. Additionally, Nguyen et al. demonstrated that Drosophila fed with boiled mature silkworm powder (BMSPF) had a significantly longer healthspan and lifespan compared to those fed with normal food [30,31]. BMSPF also showed greater resistance to Parkinson’s disease symptoms induced by rotenone, suggesting its potential in extending healthspan and preventing Parkinson’s disease progression [31]. In a scopolamine-induced amnesia rodent model, mice supplemented with SMSP exhibited enhanced mitochondrial function in the brain, which may provide a molecular basis for amnesia suppression [32]. Furthermore, Kim et al. reported that oral administration of SMSP to HRM-2 melanin-possessing hairless mice significantly reduced UVB-induced abnormal pigmentation and demonstrated potential anti-melanogenic efficacy [33].

Despite the promising potential of SMSP as a functional health food, its use is currently limited to food applications and has not yet expanded into the health supplement market. To facilitate this expansion, robust foundational research and clinical trials are necessary. Clinical trials are crucial for thoroughly evaluating SMSP’s effects on liver function improvement, Parkinson’s disease prevention, skin health, and other potential benefits. The toxicity study of SMSP presented in this paper is expected to be a critical piece of evidence for determining appropriate concentration levels in future SMSP-related clinical trials.

## Figures and Tables

**Figure 1 foods-13-02209-f001:**
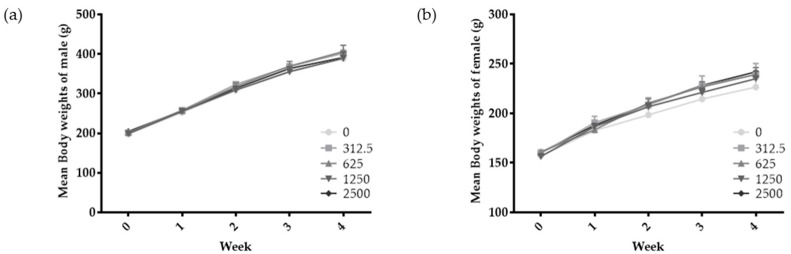
Four-week repeated-dose toxicity study: body weight changes in male (**a**) and female (**b**) rats over 4 weeks. The day of administration is designated day 1. The data are presented as mean ± S.D. (*n* = 5) and were subjected to one-way ANOVA analysis. The results showed no significant differences.

**Figure 2 foods-13-02209-f002:**
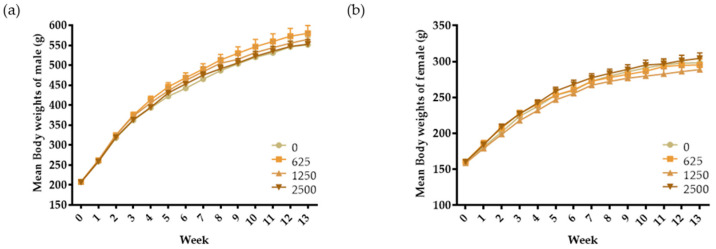
Thirteen-week repeated-dose toxicity study: body weight changes in male (**a**) and female (**b**) rats over 13 weeks. The day of administration was designated day 1. The data are presented as mean ± S.D. (*n* = 10) and were subjected to one-way ANOVA analysis. The results showed no significant differences.

**Figure 3 foods-13-02209-f003:**
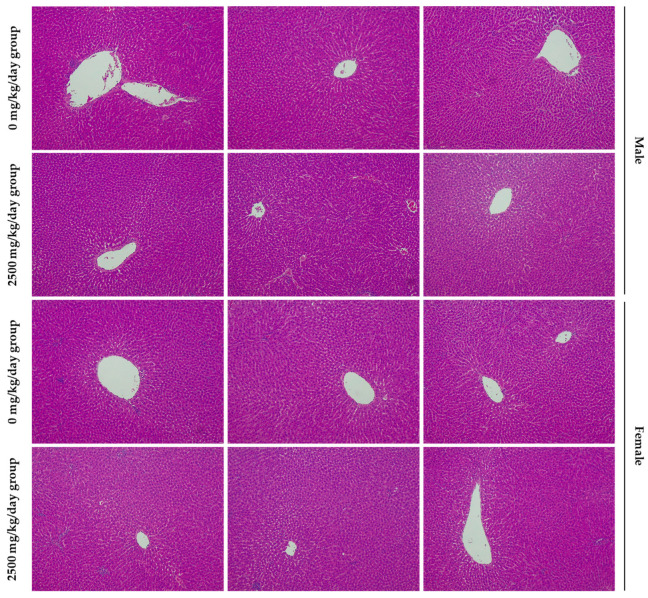
Histological examinations for liver. Samples from both the control group and the SMSP 2500 mg/kg/day group were analyzed at a magnification of 100×.

**Table 1 foods-13-02209-t001:** Thirteen-week repeated-dose toxicity study: urinalysis in male and female rats.

Test	Result	Group (mg/kg/day)			
		0	625	1200	2500
**Male**					
GLU (mg/dL)	Negative	5	5	5	5
BIL	Negative	5	5	5	5
KET (mg/dL)	Negative	1	3	5 *	0
	Trace	2	1	0	4
	15	2	1	0	1
SG	≤1.005	0	0	2	1
	1.010	5	5	3	4
pH	7.5	0	0	1	0
	8.0	1	2	3	1
	8.5	4	3	1	4
PRO (mg/dL)	Negative	2	2	4	1
	15	1	1	1	2
	30	2	1	0	2
	100	0	1	0	0
URO (EU/dL)	0.2	5	5	5	5
NIT	Negative	5	4	4	5
	Positive	0	1	1	0
BLO	Negative	4	2	2	1
	Trace	1	3	3	4
Volume (mL)		15.8 ± 4.6	14.2 ± 2.3	23.0 ± 5.5	17.4 ± 2.6
Number (N)		5	5	5	5
**Female**					
GLU (mg/dL)	Negative	5	5	5	5
BIL	Negative	5	5	5	5
KET (mg/dL)	Negative	5	4	5	4
	Trace	0	1	0	1
SG	≤1.005	0	0	1	1
	1.010	4	1	4	4
	1.015	1	4	0	0
	1.020	0	1	0	0
pH	7.0	0	1	0	0
	7.5	0	0	0	1
	8.0	0	0	2	0
	8.5	5	4	3	4
PRO (mg/dL)	Negative	3	2	5	4
	15	1	2	0	1
	30	1	1	0	0
URO (EU/dL)	0.2	4	4	5	5
	1.0	1	1	0	0
NIT	Negative	3	3	4	4
	Positive	2	2	1	1
BLO	Negative	5	5	5	4
	Small	0	0	0	1
Volume (mL)		12.2 ± 2.3	10.2 ± 4.8	10.2 ± 3.6	12.2 ± 2.3
Number (N)		5	5	5	5

GLU, glucose; BIL, bilirubin; KET, ketone body; SG, specific gravity; PRO, protein; URO, urobilinogen; NIT, nitrite; BLO, occult blood. * Significant difference at *p* < 0.05 levels compared with the control group (*n* = 5).

**Table 2 foods-13-02209-t002:** Thirteen-week repeated-dose toxicity study: hematological values in male and female rats.

Parameters	Groups (mg/kg/day)			
	0	625	1250	2500
**Male**				
RBC (10^6^/μL)	8.69 ± 0.40	8.50 ± 0.32	8.73 ± 0.30	8.33 ± 0.41
HGB (g/dL)	14.9 ± 0.6	14.7 ± 0.4	15.1 ± 0.6	14.3 ± 0.4
HCT (%)	46.4 ± 2.0	46.3 ± 1.4	47.2 ± 1.7	45.0 ± 1.5
MCV (fL)	53.4 ± 1.1	54.4 ± 1.2	54.1 ± 1.9	54.2 ± 1.7
MCH (pg)	17.1 ± 0.4	17.3 ± 0.6	17.3 ± 0.7	17.2 ± 0.5
MCHC (g/dL)	32.0 ± 0.4	31.8 ± 0.6	32.0 ± 0.4	31.8 ± 0.4
PLT (10^3^/μL)	918.0 ± 85.0	869.1 ± 100.0	866.2 ± 57.2	961.4 ± 114.5
WBC (10^3^/μL)	9.52 ± 2.58	9.08 ± 3.00	9.70 ± 1.94	10.14 ± 1.76
NEU (%)	22.2 ± 8.3	21.0 ± 9.6	24.2 ± 8.2	18.9 ± 7.2
LYM (%)	72.8 ± 8.6	74.3 ± 10.1	71.2 ± 8.2	75.9 ± 7.0
MONO (%)	3.29 ± 0.89	2.86 ± 1.30	2.71 ± 0.51	3.26 ± 0.77
EOS (%)	0.99 ± 0.25	1.11 ± 0.37	1.18 ± 0.36	1.06 ± 0.42
BASO (%)	0.20 ± 0.09	0.23 ± 0.05	0.18 ± 0.09	0.21 ± 0.06
LUC (%)	0.52 ± 0.21	0.49 ± 0.28	0.55 ± 0.24	0.63 ± 0.19
**Female**				
RBC (10^6^/μL)	7.78 ± 0.25	7.72 ± 0.25	7.71 ± 0.35	7.73 ± 0.42
HGB (g/dL)	14.3 ± 0.5	14.2 ± 0.5	14.2 ± 0.5	14.1 ± 0.3
HCT (%)	43.6 ± 1.5	43.2 ± 1.8	43.3 ± 1.3	42.8 ± 1.7
MCV (fL)	56.0 ± 1.3	56.0 ± 1.4	56.2 ± 1.7	55.4 ± 2.1
MCH (pg)	18.3 ± 0.4	18.4 ± 0.5	18.4 ± 0.6	18.3 ± 0.9
MCHC (g/dL)	32.7 ± 0.5	32.9 ± 0.6	32.8 ± 0.4	32.9 ± 0.8
PLT (10^3^/μL)	945.0 ± 92.1	919.0 ± 126.0	938.4 ± 108.5	916.4 ± 95.0
WBC (10^3^/μL)	5.35 ± 1.57	4.91 ± 1.25	6.03 ± 1.87	6.09 ± 1.09
NEU (%)	16.0 ± 4.5	16.2 ± 6.3	17.7 ± 5.9	15.8 ± 4.6
LYM (%)	78.0 ± 4.4	78.4 ± 6.8	77.0 ± 6.5	79.4 ± 4.7
MONO (%)	3.47 ± 0.87	2.85 ± 0.88	2.94 ± 0.83	2.76 ± 0.94
EOS (%)	1.76 ± 0.51	1.55 ± 0.46	1.49 ± 0.54	1.37 ± 0.49
BASO (%)	0.18 ± 0.06	0.11 ± 0.07	0.14 ± 0.08	0.13 ± 0.09
LUC (%)	0.55 ± 0.15	0.77 ± 0.34	0.65 ± 0.26	0.51 ± 0.19

RBC, red blood cell; HGB, hemoglobin; HCT, hematocrit; MCV, mean corpuscular volume; MCH, mean corpuscular hemoglobin; MCHC, mean corpuscular hemoglobin concentration; PLT, platelet; MPV, mean platelet volume; WBC, white blood cell; NEU, neutrophils; LYM, lymphocytes; MONO, monocytes; EOS, eosinophil; BASO, basophils; LUC, large unstained cells. Data are expressed as Mean ± S.D. (*n* = 9~10).

**Table 3 foods-13-02209-t003:** Thirteen-week repeated-dose toxicity study: Serum biochemical values in male and female rats.

Parameters	Groups (mg/kg/day)			
	0	625	1250	2500
**Male**				
AST (U/L)	103.6 ± 63.1	80.2 ± 13.0	80.9 ± 16.6	98.2 ± 21.2
ALT (U/L)	45.8 ± 36.7	34.0 ± 6.4	33.8 ± 8.6	36.2 ± 6.4
ALP (U/L)	92.1 ± 19.0	85.9 ± 19.3	92.7 ± 25.1	80.6 ± 12.0
CPK (U/L)	113.6 ± 37.6	104.6 ± 28.2	111.0 ± 37.4	213.4 ± 133.6
TBIL (mg/dL)	0.153 ± 0.042	0.150 ± 0.031	0.142 ± 0.022	0.153 ± 0.019
GLU (mg/dL)	143.0 ± 16.5	133.3 ± 10.8	139.6 ± 11.3	139.9 ± 17.1
TCHO (mg/dL)	69.8 ± 12.3	84.5 ± 22.9	70.1 ± 22.1	66.9 ± 17.1
TG (mg/dL)	49.9 ± 22.0	61.5 ± 35.6	69.5 ± 24.9	55.5 ± 26.1
TP (g/dL)	6.12 ± 0.21	6.13 ± 0.33	6.16 ± 0.25	6.09 ± 0.30
ALB (g/dL)	2.95 ± 0.08	2.97 ± 0.12	2.99 ± 0.12	2.95 ± 0.14
BUN (mg/dL)	12.2 ± 1.4	12.6 ± 1.9	12.1 ± 1.2	12.8 ± 1.7
CRE (mg/dL)	0.36 ± 0.02	0.41 ± 0.05	0.38 ± 0.03	0.41 ± 0.03
IP (mg/dL)	6.00 ± 0.28	5.93 ± 0.49	5.52 ± 0.19	5.97 ± 0.42
Ca^2+^ (mg/dL)	9.69 ± 0.18	9.73 ± 0.30	9.66 ± 0.25	9.53 ± 0.23
Na^+^ (mmol/L)	133.0 ± 0.8	134.2 ± 0.9	134.4 ± 1.9	133.8 ± 0.9
K^+^ (mmol/L)	4.12 ± 0.21	4.00 ± 0.29	3.94 ± 0.21	4.16 ± 0.31
Cl^−^ (mmol/L)	98.2 ± 0.9	98.9 ± 1.6	98.5 ± 1.6	97.5 ± 1.4
**Female**				
AST (U/L)	80.9 ± 27.9	79.9 ± 13.0	90.0 ± 19.0	89.0 ± 25.9
ALT (U/L)	27.3 ± 8.5	29.4 ± 6.4	31.9 ± 10.9	30.4 ± 13.0
ALP (U/L)	45.2 ± 10.8	37.5 ± 9.9	39.9 ± 13.4	41.9 ± 9.3
CPK (U/L)	143.3 ± 92.3	139.7 ± 94.8	97.5 ± 33.9	141.9 ± 64.0
TBIL (mg/dL)	0.181 ± 0.02	0.189 ± 0.03	0.224 ± 0.022	0.223 ± 0.097
GLU (mg/dL)	129.6 ± 9.9	124.3 ± 10.8	133.2 ± 15.8	129.1 ± 9.8
TCHO (mg/dL)	84.6 ± 9.5	83.7 ± 20.6	86.8 ± 18.8	84.7 ± 19.8
TG (mg/dL)	42.6 ± 14.0	47.5 ± 17.8	39.0 ± 7.1	35.6 ± 7.4
TP (g/dL)	6.75 ± 0.37	6.72 ± 0.46	6.75 ± 0.38	6.62 ± 0.43
ALB (g/dL)	3.58 ± 0.25	3.53 ± 0.31	3.59 ± 0.23	3.53 ± 0.25
BUN (mg/dL)	14.1 ± 1.3	15.5 ± 2.6	15.7 ± 2.8	15.2 ± 4.4
CRE (mg/dL)	0.45 ± 0.02	0.47 ± 0.06	0.49 ± 0.05	0.47 ± 0.06
IP (mg/dL)	4.67 ± 0.40	4.89 ± 0.74	4.82 ± 0.56	4.95 ± 0.42
Ca^2+^ (mg/dL)	10.04 ± 0.29	9.99 ± 0.26	10.02 ± 0.25	9.77 ± 0.31
Na^+^ (mmol/L)	141.0 ± 0.9	140.2 ± 1.0	140.3 ± 1.0	140.0 ± 1.5
K^+^ (mmol/L)	4.06 ± 0.25	4.06 ± 0.26	3.90 ± 0.18	3.92 ± 0.32
Cl^−^ (mmol/L)	105.8 ± 1.2	104.7 ± 1.2	105.5 ± 1.4	104.4 ± 1.8

AST, aspartate aminotransferase; ALT, alanine aminotransferase; ALP, alkaline phosphatase; CPK, creatine phosphokinase; TBIL, serum total bilirubin; GLU, glucose; TCHO, total cholesterol; TG, triglycerides; TP, total protein; ALB, albumin; BUN, blood urea nitrogen; CRE, creatine; IP, inorganic phosphorus. Data are expressed as Mean ± S.D. (*n* = 9~10).

**Table 4 foods-13-02209-t004:** Thirteen-week repeated-dose toxicity study: relative organ weights in male and female rats.

Organs	Groups (mg/kg/day)			
	0	625	1250	2500
**Male**				
Body weight	551.25 ± 57.11	580.07 ± 61.42	564.98 ± 56.34	552.79 ± 36.52
Adrenal gland L	0.032 ± 0.004	0.031 ± 0.003	0.032 ± 0.005	0.033 ± 0.005
Adrenal gland R	0.023 ± 0.003	0.030 ± 0.004	0.029 ± 0.004	0.031 ± 0.006
Thymus	0.314 ± 0.089	0.306 ± 0.058	0.322 ± 0.087	0.292 ± 0.041
Prostate gland	0.691 ± 0.122	0.589 ± 0.134	0.741 ± 0.193	0.639 ± 0.106
Testis L	1.805 ± 0.156	1.939 ± 0.175	1.946 ± 0.201	1.840 ± 0.241
Testis R	1.679 ± 0.374	1.919 ± 0.187	1.938 ± 0.172	1.836 ± 0.225
Epididymis L	0.681 ± 0.103	0.754 ± 0.062	0.765 ± 0.073	0.705 ± 0.075
Epididymis R	0.674 ± 0.166	0.778 ± 0.069	0.807 ± 0.071	0.707 ± 0.069
Spleen	1.000 ± 0.312	0.904 ± 0.121	0.894 ± 0.161	0.827 ± 0.121
Kidney L	1.619 ± 0.153	1.695 ± 0.158	1.613 ± 0.138	1.628 ± 0.141
Kidney R	1.536 ± 0.269	1.673 ± 0.154	1.621 ± 0.120	1.633 ± 0.126
Heart	1.570 ± 0.146	1.714 ± 0.180	1.624 ± 0.130	1.579 ± 0.171
Lung	1.700 ± 0.191	1.843 ± 0.169	1.776 ± 0.096	1.712 ± 0.091
Brain	2.193 ± 0.100	2.157 ± 0.072	2.128 ± 0.091	2.206 ± 0.050
Liver	12.78 ± 1.682	13.67 ± 2.214	13.91 ± 1.829	13.38 ± 1.277
**Female**				
Body weight	298.29 ± 21.23	295.24 ± 26.30	288.76 ± 21.52	304.44 ± 24.09
Adrenal gland L	0.035 ± 0.004	0.037 ± 0.006	0.035 ± 0.005	0.039 ± 0.005
Adrenal gland R	0.034 ± 0.004	0.037 ± 0.006	0.033 ± 0.006	0.038 ± 0.005
Thymus	0.239 ± 0.026	0.289 ± 0.108	0.238 ± 0.060	0.322 ± 0.099
Ovary L	0.047 ± 0.006	0.042 ± 0.011	0.040 ± 0.009	0.045 ± 0.009
Ovary R	0.042 ± 0.006	0.038 ± 0.007	0.040 ± 0.011	0.044 ± 0.010
Uterus (with cervix)	0.778 ± 0.243	0.656 ± 0.207	0.727 ± 0.259	0.622 ± 0.170
Spleen	0.568 ± 0.063	0.526 ± 0.093	0.542 ± 0.097	0.627 ± 0.162
Kidney L	0.959 ± 0.059	0.928 ± 0.068	0.930 ± 0.120	0.995 ± 0.107
Kidney R	0.953 ± 0.061	0.936 ± 0.082	0.945 ± 0.103	1.021 ± 0.119
Heart	1.045 ± 0.071	1.050 ± 0.105	1.005 ± 0.051	1.041 ± 0.074
Lung	1.332 ± 0.095	1.310 ± 0.138	1.299 ± 0.104	1.318 ± 0.102
Brain	2.051 ± 0.105	2.006 ± 0.122	1.988 ± 0.139	2.066 ± 0.089
Liver	7.465 ± 0.461	7.452 ± 0.713	7.276 ± 0.573	7.605 ± 0.719

Data are expressed as Mean ± S.D. (*n* = 9~10).

## Data Availability

The original contributions presented in the study are included in the article, further inquiries can be directed to the corresponding author.

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
