# Peer review of "Toxicological Evaluation Verifies the Safety of Oral Administration of Steamed Mature Silkworm Powder in Rats"

_foods, 2024, doi:10.3390/foods13142209_

Round 1

Reviewer 1 Report

Comments and Suggestions for Authors

The manuscript presents interesting results regarding the toxicological assessment for a silkworm powder. I have a few suggestions:

1. The title could be added with information regarding the main finding of the study.

2. Some scientific names are not in italic (example: line 44)

3. Are there any international legislations regarding using silkworm powder to date? Maybe this information could be provided in the introduction for if there aren't any, the results from the study can present more data to prove the safety of use of it.

4. The discussion part seems to lack a deeper comparison with other studies. Are there similar studies that can be compared to this? How are the safety studies with other insect species being conducted? And have they proved safety for consumption as well? How reliable is the conducted toxicity test?

5. Is it possible to run statistical analyses with the data presented for tables 2 and 3? I believe it would be a way to consolidate the hypothesis further.

6. Which other analyses and studies can be done from now on? Considering the results found? How relevant are the findings for the scientific community? I believe these information could be added to the conclusion, as it is lacking depth.

7. Some references are a bit dated and could be replaced with more recent ones.

Comments on the Quality of English Language

English quality proved to be good enough for publishing.

Author Response

The manuscript presents interesting results regarding the toxicological assessment for a silkworm powder. I have a few suggestions:

Comments 1: The title could be added with information regarding the main finding of the study.

Response 1: We have revised the title according to your suggestion as follows. Thanks.

“Toxicological Evaluation Verifies the Safety of Oral Administration of Steamed Mature Silkworm Powder in Rats”

Comments 2: Some scientific names are not in italic (example: line 44)

Response 2: According to your suggestion, we have revised all instances of Bombyx mori to be italicized. Thanks.

Comments 3: Are there any international legislations regarding using silkworm powder to date? Maybe this information could be provided in the introduction for if there aren't any, the results from the study can present more data to prove the safety of use of it.

Response 3: To date, clear international laws and regulations regarding the use of silkworm powder have not been established. However, regulatory agencies in individual countries set regulations for health supplements, requiring stability evaluations before these products can be released on the market. As you suggested, we have incorporated this information into the Introduction of the revised paper as follows (line 68).

“Currently, there are no international legislations specifically regulating the use of silkworm powder in food products. This study aims to provide comprehensive data supporting the safety of silkworm powder for consumption, contributing valuable information to the field and potentially aiding in the establishment of future regulations.”

Comments 4-1: The discussion part seems to lack a deeper comparison with other studies. Are there similar studies that can be compared to this?

Response 4-1: Thank you for your comments. A study similar to our study, and we have added this information to the Discussion section as follows (line 284).

“In comparison, a toxicity test using silkworm extract powder (SEP) reported in 2013 established a NOAEL of 2000 mg/kg BW/day. This SEP, derived from 5th instar 3rd day larvae, contained significant amounts of 1-deoxynojirimycin (DNJ), a hypoglycemic component [22]. In contrast, SMSP produced through steaming and grinding mature silkworms, contains higher levels of beneficial ingredients such as amino acids and omega-3 fatty acids and demonstrates enhanced safety. However, SMSP has very low DNJ content, likely due to the absence of silkworm feces in the fully grown silkworms used [23]. Given their different effective ingredients and functions, SEP and SMSP can be applied based on specific needs and have the potential to complement each other in development.”

Comments 4-2: How are the safety studies with other insect species being conducted? And have they proved safety for consumption as well? How reliable is the conducted toxicity test?

Response 4-2: Thank you for your critical comments. Toxicity studies have been reported for other insects such as mealworms and grasshoppers. These insects, like silkworms, are emerging as future food sources. In Europe, dried Tenebrio molitor larvae, freeze-dried and powdered locusta migratoria, Acheta domesticus, and lesser mealworm have been approved under food regulations. The European Food Safety Authority (EFSA) conducted rigorous safety assessments, confirming their safety for consumption. This growing body of research supports the safety of insect-based foods, highlighting the importance of the toxicity study of SMSP in providing scientific evidence for future certification. This information has been added to the Discussion section of the revised paper as follows (line 293).

“Toxicity studies have also been reported for other insects such as mealworms and grasshoppers as well as silkworm [24,25]. Like silkworms, these insects are gaining recognition as future food resources. Their toxicity and allergic reactions when used as food or dietary supplements have been analyzed. Specifically, in Europe, dried Tenebrio molitor larvae [26], freeze-dried and powdered locusta migratoria [27], freeze-dried and powdered acheta domesticus [28], and frozen and freeze-dried lesser mealworm [29] have been approved under food regulations. The European Food Safety Authority (EFSA) conducted rigorous evaluations of their toxicity and allergenicity, confirming their safety for consumption. Increasing research supports the safety of insect consumption, and various insects have been found to be safe as food. Therefore, the toxicity study of SMSP in this research will provide crucial scientific evidence for future official certification.”

Comments 5: Is it possible to run statistical analyses with the data presented for tables 2 and 3? I believe it would be a way to consolidate the hypothesis further.

Response 5: Thank you for comment. In this study, statistical analyses of the experimental results were conducted using one-way ANOVA. In Table 2, the p-values for males and females were 0.998 and 1.0, respectively, and in Table 3, the p-values for males and females were 0.9874 and 0.9994, respectively, all of which were above 0.05, indicating no significant differences between groups. This research paper focuses on the toxicity assessment of SMSP. The lack of significant differences between groups in the statistical analysis of the listed data indicates that the test substance has no observed toxicity.

Comments 6: Which other analyses and studies can be done from now on? Considering the results found? How relevant are the findings for the scientific community? I believe these information could be added to the conclusion, as it is lacking depth.

Response 6: Thank you for your comments. Our research team has been conducting studies on the liver function improvement effects of SMSP for several years. Additionally, various studies using SMSP, such as its anti-melanogenic effects and memory improvement effects in Parkinson’s disease, have been reported. Based on these studies, when developing and seeking recognition for SMSP as a functional health food, clinical trials are necessary. Our toxicity study plays a crucial role in determining the dosage for these clinical trials. As your suggestion, this information has been added to the Discussion section in the revised paper as follows (line 310).

“These findings have significant implications for the development of SMSP as a novel functional health food. Specifically, they are expected to pave the way for further research into SMSP’s mechanisms and its effectiveness against various diseases. Our research team has previously reported several positive effects of SMSP, including improvements in liver function [12,21]. Additionally, Nguyen et al. demonstrated that Drosophila fed with boiled mature silkworm powder (BMSPF) had a significantly longer healthspan and lifespan compared to those fed with normal food [30,31]. BMSPF also showed greater resistance to Parkinson’s disease symptoms induced by rotenone, suggesting its potential in extending healthspan and preventing Parkinson’s disease progression [31]. In a scopolamine-induced amnesia rodent model, mice supplemented with SMSP exhibited enhanced mitochondrial function in the brain, which may provide a molecular basis for amnesia suppression [32]. Furthermore, Kim et al. reported that oral administration of SMSP to HRM-2 melanin-possessing hairless mice significantly reduced UVB-induced abnormal pigmentation and demonstrated potential anti-melanogenic efficacy [33].

Despite the promising potential of SMSP as a functional health food, its use is currently limited to food applications and has not yet expanded into the health supplement market. To facilitate this expansion, robust foundational research and clinical trials are necessary. Clinical trials are crucial for thoroughly evaluating SMSP’s effects on liver function improvement, Parkinson’s disease prevention, skin health, and other potential benefits. The toxicity study of SMSP presented in this paper is expected to be a critical piece of evidence for determining appropriate concentration levels in future SMSP-related clinical trials.”

Comments 7: Some references are a bit dated and could be replaced with more recent ones.

Response 7: Thank you for your comments. Unfortunately, some of the references used are older, but they are still the only relevant studies available on the topic, as no more recent alternatives exist. We have updated other references to more current sources where applicable.

Reviewer 2 Report

Comments and Suggestions for Authors

The study present valuable data in a very timely topic related to the use of alternative protein in the diet of animals and humans. The overall quality of the paper is high. My main remarks concern the description of the experimental animals. I find it a bit confusing. The authors state that they have used 10 males and 10 females in 4 treatment groups. One cans suggest that both sexes are included in a group. At the same time it is clearly visible from the results that the authors evaluate separately both sexes. Please describe the formation of the groups more clearly. In figure 1 the number of the animals is 5, while in figure 2 it is 10. Is it a mistake or the authors have used 5 animals per group to measure the weight? If so, please explain why? The tables do not include information about the number of the animals. Please add this.

Why have the authors considered evaluating the effect of the dose in males and females separately? It will be appropriate to apply two-way ANOVA and study the effect of the dose, sex and interaction.

Please, reform the conclusions. Thus presented they merely repeat the results. Please, outline some further prospects of the study.

Author Response

Comments 1: The study present valuable data in a very timely topic related to the use of alternative protein in the diet of animals and humans. The overall quality of the paper is high.

Response 1: Thank you so much.

Comments 2: My main remarks concern the description of the experimental animals. I find it a bit confusing. The authors state that they have used 10 males and 10 females in 4 treatment groups. One cans suggest that both sexes are included in a group. At the same time it is clearly visible from the results that the authors evaluate separately both sexes. Please describe the formation of the groups more clearly. In figure 1 the number of the animals is 5, while in figure 2 it is 10. Is it a mistake or the authors have used 5 animals per group to measure the weight? If so, please explain why?

Response 2: Thank you for your comments. The results shown in Figure 1 are from the 4-week repeated-dose toxicity study, while those in Figure 2 are from the 13-week repeated-dose toxicity study. To determine the dosage for the 13-week study, we initially conducted the 4-week study. This 4-week study included five experimental groups (0, 312.5, 625, 1250, 2500 mg/kg/day), each comprising 5 male and 5 female subjects. Detailed information can be found in the Result section 3.1, titled “4-week repeated-dose toxicity study”.

Comments 3: The tables do not include information about the number of the animals. Please add this.

Response 3: Thank you for your comments. As you noted, we have included the number of animals in the table legends (Page 7, 8, 9, 10).

Comments 4: Why have the authors considered evaluating the effect of the dose in males and females separately?

Response 4: Typically, when conducting toxicity tests, the effects of sex hormones cannot be overlooked. Therefore, the experiment was performed in both male and female animals to clearly observe these effects. Additionally, because male and female animals have different organs (e.g., testis and ovaries), we recorded organ weights separately for each sex to confirm the data presented in Table 4. Thank you.

Comments 5: It will be appropriate to apply two-way ANOVA and study the effect of the dose, sex and interaction.

Response 5: Thank you for your comments. Two-way ANOVA could be used as you suggested. However, in our experiment, we used One-way ANOVA because the aim was not to compare differences between male and female animals, but rather to assess toxicity within each sex according to dose. Two-way ANOVA is typically used to compare between sexes, but we did not use it here because the standard values for males and females differ, and our study was not intended to compare these groups directly. Therefore, we presented the results separately by gender.

Comments 6: Please, reform the conclusions. Thus presented they merely repeat the results. Please, outline some further prospects of the study.

Response 6: As your suggestion, we have deleted the Conclusion section and completely rewritten the Discussion section by adding content, including future prospects and comparison with other studies. Thank you (Page 11~12).

Reviewer 3 Report

Comments and Suggestions for Authors

This paper tests the safety of a commerical product the authors called "steamed mature silkworm powder (SMSP)." It uses 20 rats (10 male and 10 female) for three dosages and a control. [A similar sample size was used in this paper https://link.springer.com/article/10.5487/TR.2013.29.4.263 so I assume it's fine.] No evidence of hazard was found. I ultimately find the experiment trustworthy: the testing is robust, plus silkworms have been eaten for centuries and should not cause health problems in mice. This paper seems like a mandatory step in getting a novel food product approved.

That said, the paper suggests "financial conflict of interest." Are the authors sure they have nothing to declare? They don't stand to make any money from the sale of their "product?" It's fine to do so, and I'm happy if this work brings silkworm powder to market, but if so then you must declare it. If not, very well…

The discussion is mostly introduction material. Only the last paragraph is true discussion.

Other Comments

15-16 "which has demonstrated protective effects against fatty liver disease as well as gastric and liver damage caused by ethanol" That's a bold claim and hard to believe. Since this paper does not actually test this product's effect on the liver, I do not think it is appropriate to make this claim in the abstract, though it is fine for the introduction.
44 Are silkworms themselves rich in silk protein, or their silk? [in the discussion you answer this question, but it should be addressed sooner]
49-50 Reference 12 looks at silkworm excrement/frass, not silkworm. Paper 13 is about development of a silkworm model for Parkinson's, and does not in any way state that eating silkworm can treat parkinsons! That statement absolutely must be revised, and the reference deleted.
60 You say "has been shown" but for transparency's sake I think you should disclose that your lab did this study as well.
65 I don't think the need is very urgent: steamed silkworm has been eaten in East Asia for centuries. It's generally recognized as safe.

Table 1: I am having trouble understanding the "Result" column. What do these numbers mean?

241-250 Redundant with the introduction and can be deleted.
251-260 Some of this can be moved to the introduction, the rest deleted as it is redundant
261-276 This paragraph and half paragraph justify this experiment, therefore they must absolutely be moved to the introduction
277-284 Are you talking about the experiment described in the results, or the "data not shown" study from the beginning of this paragraph? If the former, then this is discussion. if the latter, then it's introduction.
295-299 This conclusion is not necessary: your actual "discussion" is only one paragraph long.
311-314 Why is this not a normal Table?

There are very few references, only 22. Of these 9 are generic "insects are future food" articles, and 7 or so are the authors' own papers.

Comments on the Quality of English Language

Throughout the paper, be sure to italicize scientific names like Bombyx mori.

291 delete "our"

352 Typo in the first author's initial

Author Response

Comments 1: This paper tests the safety of a commercial product the authors called "steamed mature silkworm powder (SMSP)." It uses 20 rats (10 male and 10 female) for three dosages and a control. [A similar sample size was used in this paper https://link.springer.com/article/10.5487/TR.2013.29.4.263 so I assume it's fine.] No evidence of hazard was found. I ultimately find the experiment trustworthy: the testing is robust, plus silkworms have been eaten for centuries and should not cause health problems in mice. This paper seems like a mandatory step in getting a novel food product approved.

Response 1: Thank you so much.

Comments 2: That said, the paper suggests "financial conflict of interest." Are the authors sure they have nothing to declare? They don't stand to make any money from the sale of their "product?" It's fine to do so, and I'm happy if this work brings silkworm powder to market, but if so then you must declare it. If not, very well…

Response 2: Thank you for your concern. We conducted this study as part of a government research project, in which we received silkworm powder samples from a company for toxicity testing. We have no financial conflicts of interest to declare, as our research is not aimed at generating profit.

Comments 3: The discussion is mostly introduction material. Only the last paragraph is true discussion.

Response 3: Thank you for your feedback. Following your suggestion, we deleted the introduction section from the discussion and added information on various research directions for further studies and potential applications involving silkworms (Page 11~12). 

Other Comments

Comments 4: 15-16 "which has demonstrated protective effects against fatty liver disease as well as gastric and liver damage caused by ethanol" That's a bold claim and hard to believe. Since this paper does not actually test this product's effect on the liver, I do not think it is appropriate to make this claim in the abstract, though it is fine for the introduction.

Response 4: Thank you. We have deleted and revised that part of Abstract based on your feedback (Page 1, line 13~15) as follows. The corrected parts are indicated in italics.

To address this, we have developed a steamed and freeze-dried mature silkworm larva powder (SMSP), and it is essential to investigate its potential toxicity and food safety for further studies and applications.”

Comments 5: 44 Are silkworms themselves rich in silk protein, or their silk? [in the discussion you answer this question, but it should be addressed sooner]

Response 5: Following your suggestion, we moved the section from the Discussion to the Introduction to improve the clarity of the paper (Page 2, line 60) as follows. Thank you for the suggestion.

“When comparing non-steamed silkworm with SMSP, the crude protein content was 68.7% for steamed silkworms and 62.4% for non-steamed silkworms, indicating a significant difference [7]. Additionally, the levels of serine (1.1 times), glycine (1.3 times), alanine (1.3 times), and tyrosine (1.2 times) were higher in SMSP than in non-steamed silkworm [7]. There were no differences in fatty acids, vitamins, dietary fiber, or moisture content, but the protein content, making up 69 ~ 72% of the general components of SMSP, significantly contributes to its functionality [7]. Therefore, toxicity tests using SMSP were essential to assess whether such high protein intake could affect kidney and liver function or cause other disorders.

Comments 6: 49-50 Reference 12 looks at silkworm excrement/frass, not silkworm. Paper 13 is about development of a silkworm model for Parkinson's, and does not in any way state that eating silkworm can treat parkinsons! That statement absolutely must be revised, and the reference deleted.

Response 6: Thank you. As your comment, we deleted that part and replaced it with content related to silkworm as follows (Page 2).

“According to recent studies, silkworms have been shown to provide significant health benefits, including positive effects on liver disease [12], pancreatic protective effect [13], anti-hypertensive [14], hypoglycemic effect [15,16], anti-atopic dermatitis effect [17], preventive effect of memory impairment [18], and antioxidant activity [19].”

Comments 7: 60 You say "has been shown" but for transparency's sake I think you should disclose that your lab did this study as well.

Response 7: This was done by our research team, and we have revised the expression as you suggested. Thank you for comment as follows (Page 2, line 78). The corrected parts are indicated in italics.

“Additionally, our previous study has shown that SMSP mitigates hepatocellular carcinogenesis and hepatic fibrosis by inhibiting transforming growth factor (TGF)-β activity and the phosphorylation of signal transducer and activator of transcription (STAT)-3 signaling in diethylnitrosamine-induced hepatocellular carcinoma rat models”

Comments 8: 65 I don't think the need is very urgent: steamed silkworm has been eaten in East Asia for centuries. It's generally recognized as safe.

Response 8: As your suggestion, we have removed the term “urgent”. Thank you.

Comments 9: Table 1: I am having trouble understanding the "Result" column. What do these numbers mean?

Response 9: We analyzed urine samples from 5 animals in both male and female groups, and the table includes the number of animals (N) for each measured value. It seems the explanation may not have been clear enough. To improve understanding, we have added a brief description of the table to the Result section (Page 5).

“Table 1 shows the results of analyzing five urine samples from each group, with each number representing the corresponding number of animals.”

Additionally, to make the table easier to understand as your suggestion, we have deleted Appendix Table 1 and incorporated its content into Table 1 in the revised paper.

Comments 10: 241-250 Redundant with the introduction and can be deleted.

Response 10: As your suggestion, we have removed the section because it was redundant with the introduction.

Comments 11: 251-260 Some of this can be moved to the introduction, the rest deleted as it is redundant

Response 11: Thank you for your comment. As you opinion, we deleted the redundant part from the Discussion, moved it to the Introduction, and revised it accordingly (Page 1~2).

Comments 12: 261-276 This paragraph and half paragraph justify this experiment, therefore they must absolutely be moved to the introduction

Response 12: Following your suggestion, we moved the section from the Discussion to the Introduction to improve the clarity of the paper (Page 2). Thank you.

Comments 13: 277-284 Are you talking about the experiment described in the results, or the "data not shown" study from the beginning of this paragraph? If the former, then this is discussion. if the latter, then it's introduction.

Response 13: Thank you. Since parts 277 to 284 discuss the experiments conducted in this study, we have retained this section in the Discussion, as it aligns with your suggestion.

Comments 14: 295-299 This conclusion is not necessary: your actual "discussion" is only one paragraph long.

Response 14: As your suggestion, we have deleted the Conclusion section and completely rewritten the Discussion section by adding content, including future prospects and comparison with other studies. Thank you (Page 11~12).

Comments 15: 311-314 Why is this not a normal Table?

Response 15: As your comments and to make the Table 1 easier to understand, we have deleted Appendix Table 1 and incorporated its content into Table 1 in the revised paper. Thanks.

Comments 16: There are very few references, only 22. Of these 9 are generic "insects are future food" articles, and 7 or so are the authors' own papers.

Response 16: The authors of this research paper have been studying SMSP in various models for many years. In this paper, we cited our previous research to explain the use of SMSP, but we did not realize that it resulted in excessive self-citation. Thank you for your critical comments. In the revised paper, we have included studied on SMSP or silkworms by referring to other sources.

Comments on the Quality of English Language

Comments 17: Throughout the paper, be sure to italicize scientific names like Bombyx mori.

Response 17: In response to your suggestion, we have revised all instances of Bombyx mori to italicize them.

Comments 18: 291 delete "our"

Response 18: As your opinion, we deleted. Thanks.

Comments 19: 352 Typo in the first author's initial

Response 19: As your suggestion, we have corrected the typo the references in revised manuscript. Thanks.